# Structure-Phase Transformations in the Course of Solid-State Mechanical Alloying of High-Nitrogen Chromium-Manganese Steels

**Kirill Lyashkov [1], Valery Shabashov [2,\*], Andrey Zamatovskii [2], Kirill Kozlov [2], Natalya Kataeva [2], Evgenii Novikov [2] and Yurii Ustyugov [2]**

[1] Institute of Engineering Science, Ural Branch, Russian Academy of Sciences, 620108 Ekaterinburg, Russia; lyashkov@imp.uran.ru

[2] Mikheev Institute of Metal Physics, Ural Branch, Russian Academy of Sciences, 620108 Ekaterinburg, Russia; zamatovsky@imp.uran.ru (A.Z.); kozlov@imp.uran.ru (K.K.); kataeva@imp.uran.ru (N.K.); evg_nov@mail.ru (E.N.); ustyugov@imp.uran.ru (Y.U.)

\* Correspondence: shabashov@imp.uran.ru

**Abstract:** The solid-state mechanical alloying (MA) of high-nitrogen chromium-manganese austenite steel—MA in a planetary ball mill, —was studied by methods of Mössbauer spectroscopy and transmission electron microscopy (TEM). In the capacity of a material for the alloying we used mixtures of the binary Fe–Mn and Fe–Cr alloys with the nitrides CrN ($Cr_2N$) and $Mn_2N$. It is shown that ball milling of the mixtures has led to the occurrence of the $\alpha \rightarrow \gamma$ transitions being accompanied by the (i) formation of the solid solutions supersaturated with nitrogen and by (ii) their decomposition with the formation of secondary nitrides. The austenite formed by the ball milling and subsequent annealing at 700–800 °C, was a submicrocrystalline one that contained secondary nano-sized crystalline CrN ($Cr_2N$) nitrides. It has been established that using the nitride $Mn_2N$ as nitrogen-containing addition is more preferable for the formation and stabilization of austenite—in the course of the MA and subsequent annealing—because of the formation of the concentration-inhomogeneous regions of $\gamma$ phase enriched with austenite-forming low-mobile manganese.

**Keywords:** austenitic alloys; mechanical alloying; high-nitrogen steels; atomic redistribution; point defects; Mössbauer spectroscopy



## 1. Introduction

In the modern technology of production of the materials with specific (improved) functional properties a great emphasis is placed on the design of the high-nitrogen steels, alloys, and composites [1–3]. This is due to the (i) need for economically alloyed steels and (ii) feasibility of substitution of nitrogen for the high-cost nickel and manganese. The current technology for melting high-nitrogen steels under nitrogen pressure requires expensive equipment and high energy expenditure. In the capacity of an alternative and cheaper method for producing high-nitrogen steels, today powder metallurgy methods using mechanical alloying are proposed. In most studies on the mechanical alloying (MA) of high-nitrogen steels, the metal powder was saturated with nitrogen from the gas phase in a flowing atmosphere of nitrogen or ammonia [4–13]. In these works, the authors usually employed vibrationally assisted ball milling with long-term processing times (100 h or more). In the works [14–20], for the purpose of nitriding Fe-based steels, the authors were the first to have employed the method of solid-phase MA (i.e., MA of materials in a solid state). This method does not require additional gas equipment to create a flowing nitrogen-containing atmosphere in the mill vessel. In addition, the method does not require such a long time of mechanical processing and allows for one to use planetary-type ball mills. To the basis of the approach in [15–18] their authors have placed the cyclically

evaluating deformation-induced phase transitions «dissolution—precipitation» of disperse particles of the prime low-stable nitrides (of the types CrN and $Fe_4N$) located in metallic matrices. In result of the deformation-induced dissolution of prime (initially present) nitrides, in the metallic matrix of alloys one can observe the formation of both the nano-structured solid solutions supersaturated with nitrogen and highly disperse secondary nitrides. Within the framework of the proposed approach with the employment of the (i) method of severe plastic deformation (namely, by high pressure torsion in the rotating Bridgman anvils) [15,16], (ii) friction-induced external action [16], and (iii) ball milling (BM) [17–20], it became feasible to have performed alloying of the Fe–Ni–Cr–N and the Fe–Mn–Cr–N austenite. For the development of high-nitrogen steels as competitors to austenite Fe–Ni–Cr–N alloys, the alloys based on the system Fe–Cr–Mn–N were proposed in [21,22] for consideration. First experiments on the solid-state MA of Fe–Cr–Mn–N steels using BM were performed on the mixtures of the binary Fe–Mn alloy and CrN ($Cr_2N$) nitrides [17], as well as on the mixtures of the pure metals Fe, Mn, and Cr—on the one hand, and the nitrides CrN and $Mn_2N$, on the other [19].

The purpose of this work is the investigation of the possibility to obtaining nitrogen-containing austenite by means of employment of the solid-phase MA where in the capacity of the starting material a researcher uses initial mixtures of the binary alloys and nitrides together with the alloying elements that traditionally are taken as constituents of the stainless steels from the austenite class of Fe–Cr–Mn–N type. The subject of the study was the analysis of the mechanism, kinetics, and properties of the products of the MA—in the course of BM—and subsequent thermal anneals, in dependence of initial compositions of the initial matrices and nitrides. In particular, an optimum composition of the austenite to obtain was of interest from the point of view of reducing the volume of alloying manganese and chromium in the composition of MA-produced austenite.

## 2. Experimental

### 2.1. Sample Compositions and Their Treatment

For obtaining MA Fe–Cr–Mn–N alloys, in the capacity of their metallic matrix we used powders of the binary alloys with BCC crystal lattice of compositions: Fe–$X$Mn ($X$, wt% = 0, 4.0, 6.7, 8.9) and Fe–$Y$Cr ($Y$, wt% = 0, 4.7, 8.6, 14.2), and in the capacity of "donor" of the alloying element «nitrogen» there were chosen the nitrides CrN ($Cr_2N$) and $Mn_2N$, which are low-stable in conditions of deformation. We analyzed the mixtures in the series $A$ and $B$ (see Tables 1 and 2), each of which had several compositions of constituents with the varying contents of the metallic matrix and 10 or 20 wt% proportion of the nitrogen-containing addition:

$A$: Fe-$X$Mn + 20CrN ($Cr_2N$);

$B$: Fe-$Y$Cr + (10 or 20$Mn_2N$).

**Table 1.** The formula of composition, the content of alloying elements, and the quantity of austenite in the mechanical alloying (MA) alloys of series *A* after the ball milling and subsequent anneals.

| No. | Formula of Composition FeXMn + 20CrN (X in wt%) | Ball Milling for 10 h | | | Annealing after BM at 700 °C (*a*) and 800 °C (*b*) | | | | | |
|---|---|---|---|---|---|---|---|---|---|---|
| | | Quantity of Austenite γ, vol.% | Content of Nitrogen in Austenite $\gamma C_N$, wt% | Contents of Cr and Mn in the α Phase $\alpha C_{Mn+Cr}$, wt% | Quantity of Austenite γ, vol.% | | Content of Nitrogen in Austenite $\gamma C_N$, wt% | | Contents of Cr and Mn in the α Phase $\alpha C_{Mn+Cr}$, wt% | |
| | | | | | *a* | *b* | *a* | *b* | *a* | *b* |
| 1 | X = 0.0 | 6 | 3.7 | 4.0 | | | | | | |
| 2 | X = 4.0 | 7 | 1.4 | 5.2 | 0.0 | 5.0 | - | - | 2.3 | 5.0 |
| 3 | X = 6.7 | 10 | 1.1 | 7.0 | 3.0 | 75.0 | - | 0.5 | 2.2 | 4.6 |
| 4 | X = 8.9 | 36 | 1.1 | 10.0 | 30.4 | 75.0 | 0.5 | 0.1 | 1.7 | 5.2 |

**Table 2.** The formula of composition, the content of alloying elements, and the quantity of austenite in the MA alloys of series *B* after the ball milling and subsequent anneals.

| No. | Formula of Composition FeYCr + 20Mn₂N (Y in wt%) | Ball Milling for 10 h | | | Annealing after BM at 700 °C (*a*) and 800 °C (*b*) | | | | | |
|---|---|---|---|---|---|---|---|---|---|---|
| | | Quantity of Austenite γ, vol.% | Content of Nitrogen in Austenite $\gamma C_N$, wt% | Contents of Cr and Mn in the α Phase $\alpha C_{Mn+Cr}$, wt% | Quantity of Austenite γ, vol.% | | Content of Nitrogen in Austenite $\gamma C_N$, wt% | | Contents of Cr and Mn in the α Phase $\alpha C_{Mn+Cr}$, wt% | |
| | | | | | *a* | *b* | *a* | *b* | *a* | *b* |
| 1 | Y = 0.0 | 90 | 1.6 | 11.0 | 95 | 100 | - | 0.3 | - | - |
| 2 | Y = 14.2 | 95 | 2.6 | - | 100 | 100 | - | - | - | - |

| No. | Formula of Composition FeYCr + 10Mn₂N (Y in wt%) | Ball Milling for 10 h | | | Annealing after BM at 700 °C (*a*) and 800 °C (*b*) | | | | | |
|---|---|---|---|---|---|---|---|---|---|---|
| | | Quantity of Austenite γ, vol.% | Content of Nitrogen in Austenite $\gamma C_N$, wt% | Contents of Cr and Mn in the α Phase $\alpha C_{Mn+Cr}$, wt% | Quantity of Austenite γ, vol.% | | Content of Nitrogen in Austenite $\gamma C_N$, wt% | | Contents of Cr and Mn in the α Phase $\alpha C_{Mn+Cr}$, wt% | |
| | | | | | *a* | *b* | *a* | *b* | *a* | *b* |
| 3 | Y = 0.0 | 10 | 1.1 | 4.0 | 28 | 10 | 0.4 | 0.4 | 2.0 | 4.0 |
| 4 | Y = 4.7 | 24 | 1.0 | 8.7 | 24 | 80 | 0.1 | 0.1 | 2.5 | 5.5 |
| 5 | Y = 8.6 | 36 | 0.7 | 11.5 | 31 | 100 | 0.3 | 0.3 | 4.6 | - |

Element formulas of compositions were selected based on the data from phase diagrams for the states of equilibrium of stainless austenitic steels, including those doped with nitrogen [21,22]. The binary alloys were smelted, homogenized, and then filed. The nitrides were synthesized using the technology of self-propagating high-temperature synthesis [23]. According to the data of X-ray diffraction (XRD) analysis, the chromium nitrides have presented by themselves the mixture of composition 80% CrN + 20% $Cr_2N$ [18,19], and the manganese nitrides have been represented by 80% $Mn_2N$ in mixture with the nitrides that have had an enhanced content of magnesium. According to the results of chemical analysis, the synthesized magnesium nitrides have contained 9.1 wt% (28.3 at.%) nitrogen [20]. Mechanical alloying was performed with the employment of a planetary ball mill «Pulverisette-7». The speed of rotation of the ball mill platform was 800 rpm. The vessel and balls were made of high-strength ball-bearing steel containing 1.5 wt% Cr and 1.0 wt% C, iron in balance. This steel is the most stable compared to materials such as tungsten carbide and stainless steel [24]. The weight ratio for the balls (the total quantity of 15 pieces, 10 mm in diameter each) and a powder sample was 6:1. After loading the balls and powder mixtures into the vessel, air was pumped out to $10^{-3}$ mm Hg and the vessel was filled with an inert gas-argon. The duration of milling time was varied from 5 to 20 h. The temperature at the external sides of the vessels did not exceed 70 °C. The average size of the particulates of the initial powders of the alloys was about 200 μm. After termination of ball milling, part of the powders were annealed in vacuum at temperatures of 700 °C (*a*) and 800 °C (*b*) for 1 h, to specify—in a vacuum of $10^{-5}$ mm Hg. Possible contamination of the test samples with wear products was controlled by weighing the mass of vessels, balls, and powder before and after mechanical alloying. The difference in the masses of powders did not exceed 0.3–0.5 wt%. The method of estimating powder contamination by weighing the mass of the vessels, balls, and powder is not absolute, i.e., it is not free of «errors». However, in this work, (i) taking into account the same conditions of exposure of different mixtures to mechanical processing and (ii) assuming the same systematic errors (in terms of the degree of contamination), the differences in the results of mechanical alloying of mixtures can be considered reliable. According to the results of chemical analysis, in all the cases of ball milling, the oxygen concentration in the MA samples did not exceed 0.8 wt%.

## 2.2. Mössbauer and TEM Analysis of MA Samples

The Mössbauer measurements of the product results of the MA and subsequent annealing of the powders were carried out at room temperature on the spectrometer unit MS-1101 in regime of constant accelerations with the source $^{57}$Co(Rh). The spectra were calibrated using the absorbent material of α-Fe at a room temperature. Both the structure and the phase composition of an MA powder mixture were studied with the help of a transmission electron microscope JEM-200CX, with analyzing XRD patterns and dark-field images.

The Mössbauer spectra (transmission of gamma-quanta *A*, % as function of Doppler velocity *V*, mm/s) of MA samples have revealed in themselves a multicomponent structure typical of the solid solutions in both α-ferrite and γ-austenite phase states. Taking into account the complicated character of Mössbauer spectra, we used for their calculation the application-oriented soft-ware package MS Tools (borrowed from) [25]. Figures 1–5 show the experimental Mössbauer spectra and the results of their calculation. In particular, the program DISTRI was employed, which is usually used for—in the case of—locally inhomogeneous systems with a multicomponent structure and poor spectrum resolution. With its help one can improve the quality of a resolution of such spectra via restoring the distributions $p(V)$ and $p(H)$, which present by themselves the probabilities of resonance absorption represented on the scale of the Doppler velocities *V* and effective field *H*. Further, based on the (i) type of distributions $p(V)$ and $p(H)$, (ii) analysis of a priori data, and (iii) selection of the model for representing Mössbauer spectra, we made use of the soft-ware program SPECTR. For the calculating the parameters of the hyper-fine structure and partial contribution of the spectra components, we employed a standard procedure of

approximating integral spectra by a superposition of the components with a Lorentzian shape of the lines.

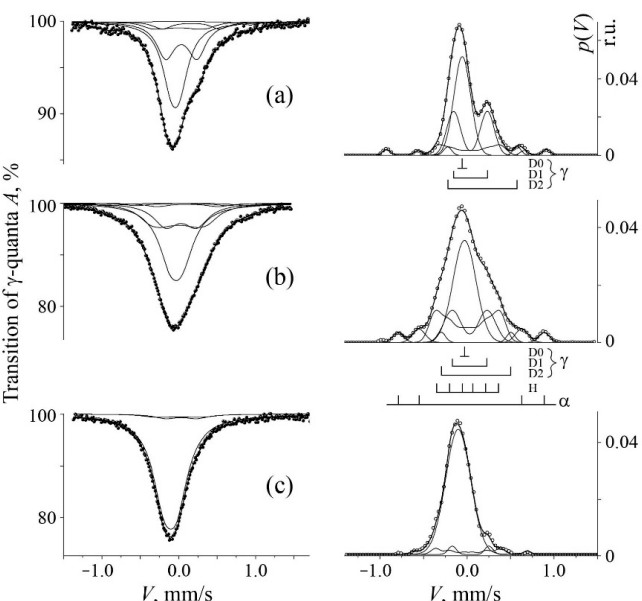

**Figure 1.** Mössbauer spectra and the distribution $p(V)$ for MA samples of Series *B*. Treatment, composition: (**a**) ball milling (BM), composition 2 (Fe–14.2Cr + 20% Mn$_2$N); (**b**) BM, composition 1 (Fe + 20% Mn$_2$N); (**c**) BM, annealing at 800 °C, 1 h, composition 1 (Fe + 20% Mn$_2$N). In the distributions $p(V)$ a "deciphering" of the model of nitrogen-containing austenite is given.

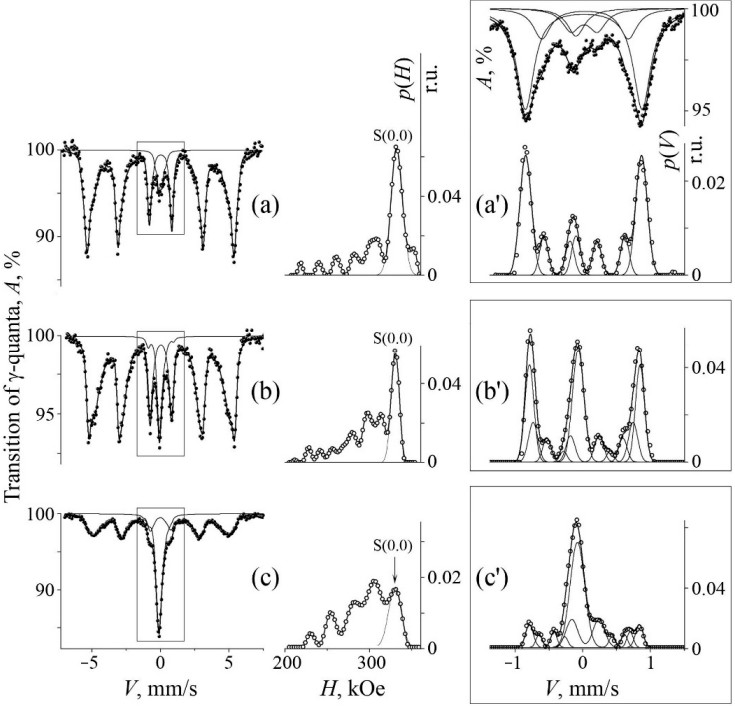

**Figure 2.** Mössbauer data of the MA samples after BM of Series *A*: (**a–c**) the Mössbauer spectra and distributions of the $p(H)$ of the sextet for the $\alpha$ phase; (**a'**) the Mössbauer spectra and distribution of the $p(V)$ of the center of the spectrum; (**b',c'**) distributions of the $p(V)$ of the center of the spectra. Treatment, composition: (**a,a'**) BM, composition 1 (Fe + 20% CrN); (**b,b'**) BM, composition 3 (Fe–6.7Mn + 20% CrN); (**c,c'**) BM, annealing at 800 °C, 1 h, composition 4 (Fe–8.9Mn + 20% CrN).

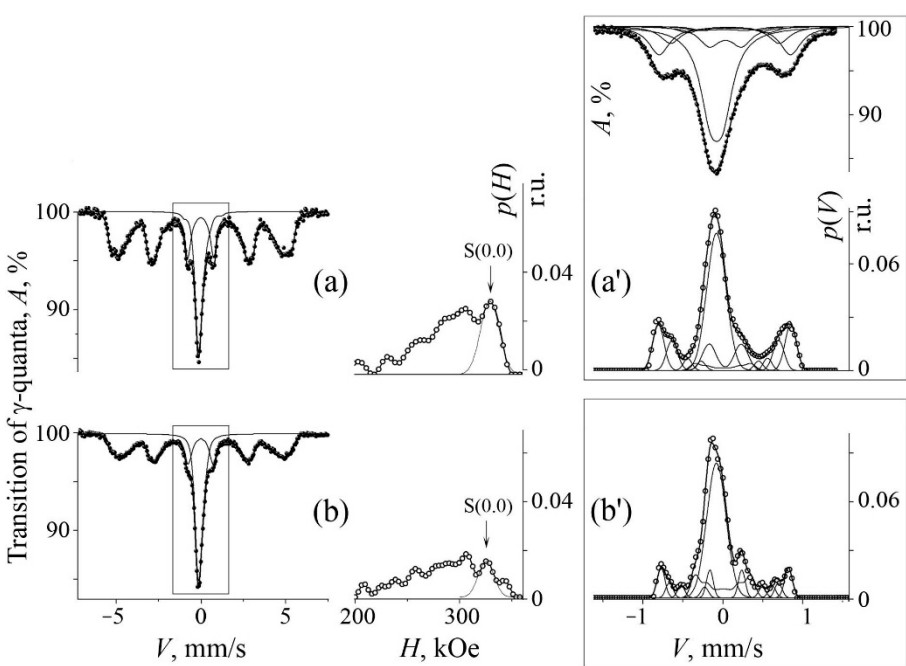

**Figure 3.** Mössbauer data of the MA samples after BM of Series *B*: (**a**,**b**) the Mössbauer spectra and distributions of the *p(H)* of the sextet for the α phase; (**a'**) the Mössbauer spectra and distribution of the *p(V)* of the center of the spectrum; (**b'**) distribution of the *p(V)* of the center of the spectrum. Treatment, composition: (**a**,**a'**) BM, composition 4 (Fe–4.7Cr + 10% Mn$_2$N); (**b**,**b'**) BM, composition 5 (Fe–8.6Cr + 10% Mn$_2$N).

For verification of the results of the mechanical alloying and modeling of the spectra of the austenite of MA alloys, special experiments were performed on the samples of the compositions 1 and 2 of the *B* series: Fe–14.2Cr + 20Mn$_2$N and Fe + 20Mn$_2$N, see Table 2. Element formulas of the special compositions in the case of mechanical alloying permitted us to compare the obtained spectra with those well-known of the alloys produced by means of traditional metallurgy [26–36]. In the composition 1 from the series *B*, in accordance with the phase diagram borrowed from [21,22], in the case of MA one can expect the realization of the spectrum for paramagnetic nitrogen austenite, whereas in the composition 2 from the series *B*—one has the right to expect the realization of the spectrum of/from γ phase with a hyper-fine magnetic structure, the formation of which being the result of an anti-ferromagnetic (AFM) ordering in the Fe–Mn alloy with large (≥20%) content of manganese [29,32,33]. The results of calculation of the *p(V)* and spectra for the MA alloys of compositions 1 and 2 of (a) *B* series are shown in Figure 1a,b. The distribution *p(V)* reveals by itself a hyper-fine structure of the spectra of/from nitrided austenite, with that the spectra restored with the help of the data over *p(V)* describe the experiment well. Indeed, the spectra of/from the MA alloy of composition 1 of *B* series, judging by their shape and hyper-fine parameters agree well with the well-known data on the spectrum of/from nitrided paramagnetic austenite [16,30,31] (see Figure 1a and Table 3). In the case of samples of composition 2 from the series *B*, the measurements at room temperature show for the spectrum of/from the alloy the realization of broadening of the central singlet, which is characteristic of the AFM ordering in the FCC Fe–Mn austenite that contains manganese in quantity equal or greater (≥) than 20 wt% [29,32,34], Figure 1b and Table 1. To confirm the AFM origin of the broadening in the spectra of/from the MA alloy of composition 2 of series *B* there serves the emergence of the narrowing of the central singlet, which is due to the (i) partial decomposition of FCC solid solution and the (ii) exit of the manganese and nitrogen atoms from the matrix into nitrides, Figure 1c. When approximating the distribution *p(V)* by Gaussian forms, the model of the spectrum can be represented by the superposition of the components—*D*(0) + *D*(1) + *D*(2) + *S*$_{AFM}$ with the parameters

given in Table 3. The selection of the Gaussian forms for the approximation of the lines and the very modeling itself are based on a priori info about the spectrum of the stainless steel [34,35] and nitrogen austenite [16,30,31]. The doublet $D(0)$ corresponds to the atoms of resonance-exhibiting iron without admixture of nitrogen, which is located in the octahedral interstitial positions—in the crystal lattice—of the first coordination shell (CS), and—by the magnitudes of the quadrupole shift and width of Gaussian forms—is close to the values of distribution (of probabilities of observation) of the electric field gradient and isomer shift for stainless steel [34,35]. The doublets $D(1)$ and $D(2)$ are of hyperfine parameters (namely, isomer shift $I_s$ and quadrupole shift $Qs$) in value close to the parameters of the doublets from intrusion of one and two (so-called dumbbell configurations) atoms of nitrogen into the octahedral interstitial positions in the crystal lattice of FCC iron [30,31]. The sextet $S_{AFM}$ with an hyperfine magnetic structure corresponds to a part of the formed $\gamma$ phase with a large ($\geq$20%) content of manganese [32,33]. The spectra and distributions $p(V)$ are presented in Figure 1, and the parameters, in Table 3. Thus, special experiments with the samples of compositions 1 and 2 in the series $B$ testify to the attainment of a desired mechanical alloying and to the possibility of modeling of spectra of MA alloys on the basis of models of Mössbauer spectra of nitrided austenite produced by traditional metallurgical methods [29–34].

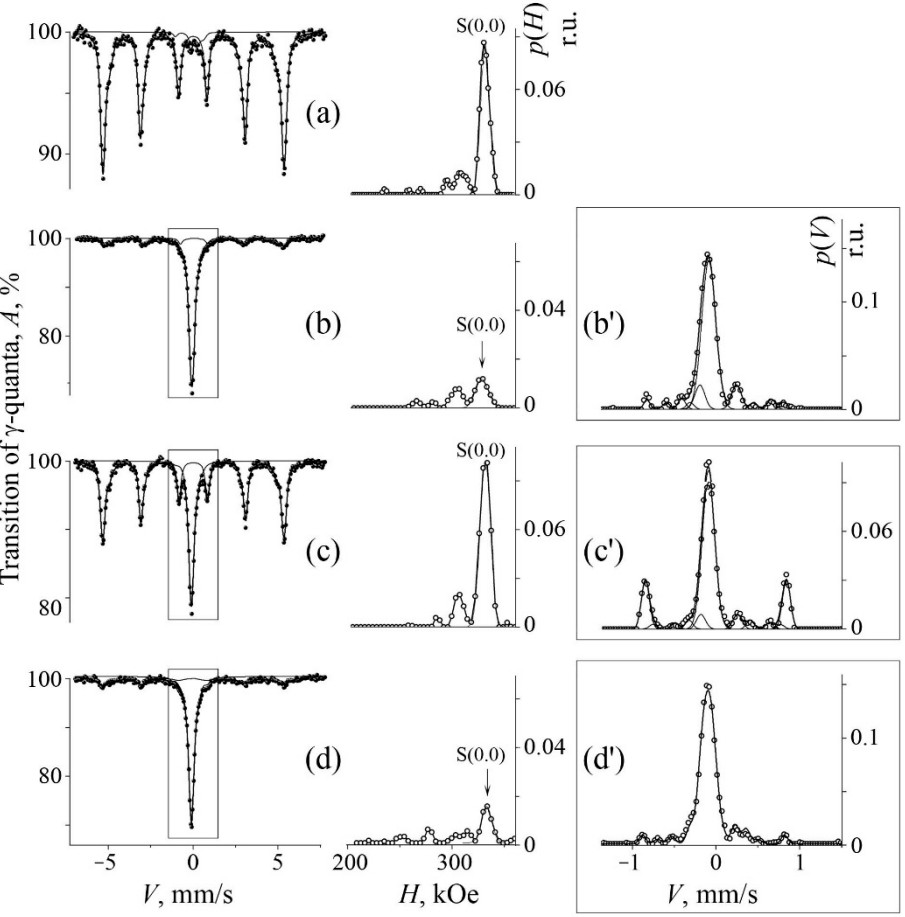

**Figure 4.** Mössbauer data of the MA samples after BM and annealing of Series $A$: (**a**–**d**) distributions of the $p(H)$ of the sextet for the $\alpha$ phase; (**b'**–**d'**) distributions of the $p(V)$ of the center of the spectra. Treatment, composition: (**a**) BM, annealing at 700 °C, 1 h, composition 3 (Fe–6.7Mn + 20% CrN); (**b,b'**) BM, annealing at 800 °C, 1 h, composition 3 (Fe–6.7Mn + 20% CrN); (**c,c'**) BM, annealing at 800 °C, 1 h, composition 4 (Fe–8.9Mn + 20% CrN); (**d,d'**) BM, annealing at 800 °C, 1 h, composition 4 (Fe–8.9Mn + 20% CrN).

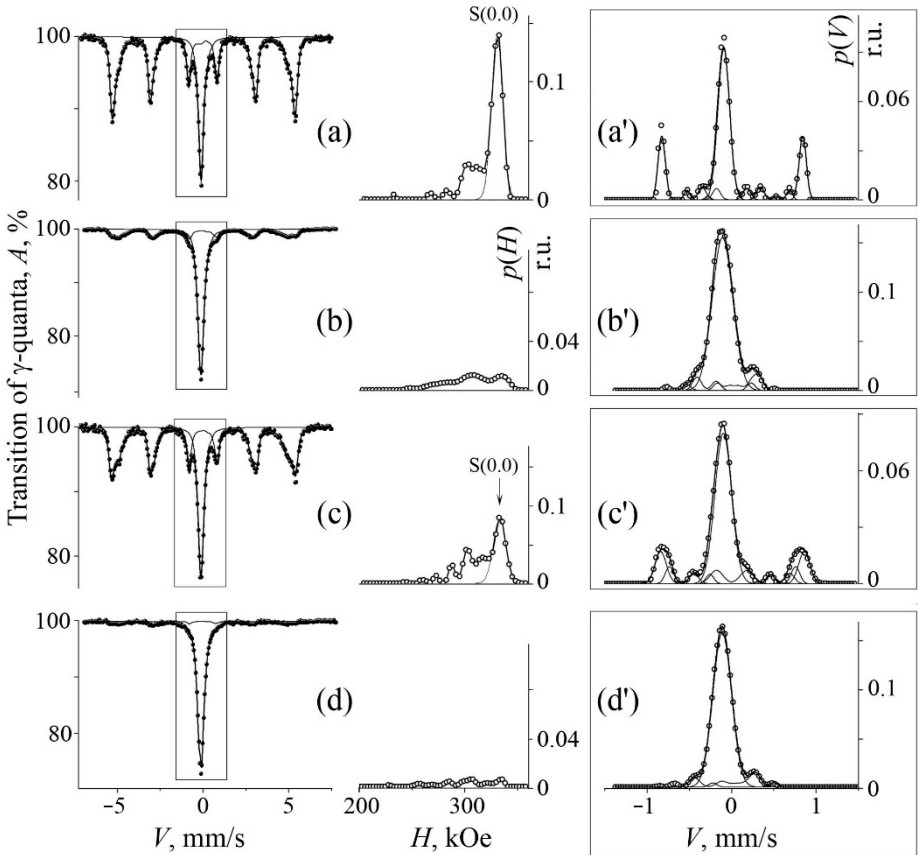

**Figure 5.** Mössbauer data of the MA samples after BM and annealing of Series *B*: (**a–d**) distributions of the *p*(*H*) of the sextet for the α phase; (**a′–d′**) distributions of the *p*(*V*) of the center of the spectra. Treatment, composition: (**a,a′**) BM, annealing at 700 °C, 1 h, composition 4 (Fe–4.7Cr + 10% Mn₂N); (**b,b′**) BM, annealing at 800 °C, 1 h, composition 4 (Fe–4.7Cr + 10% Mn₂N); (**c,c′**) BM, annealing at 700 °C, 1 h, composition 5 (Fe–8.6Cr + 10% Mn₂N); (**d,d′**) BM, annealing at 800 °C, 1 h, composition 5 (Fe–8.6Cr + 10% Mn₂N).

**Table 3.** Hyper-fine parameters of simulated spectra of the MA nitrogen-containing austenite, stainless austenite Fe–22Mn–18Cr–0.8N steel, and nitrogen-containing γ iron.

| Formula of Composition | Doublet *D*(0) | | Doublet *D*(1) | | Doublet *D*(2) | | Sextet *S*$_{AFM}$ | | | Comments and Refs |
|---|---|---|---|---|---|---|---|---|---|---|
| | *I*$_S$, mm/s ±0.01 | *Q*$_S$, mm/s ±0.01 | *I*$_S$, mm/s ±0.01 | *Q*$_S$, mm/s ±0.01 | *I*$_S$, mm/s ±0.01 | *Q*$_S$, mm/s ±0.01 | *I*$_S$, mm/s ±0.02 | *Q*$_S$, mm/s ±0.02 | *H*, kOe ±2 | |
| Fe + 20% CrN | −0.07 | 0.00 | 0.03 | 0.19 | 0.18 | 0.41 | - | - | - | Series *A*, composition 1 |
| Fe-14.2Cr + 20% Mn₂N | −0.05 | 0.14 | 0.03 | 0.20 | 0.16 | 0.36 | − | - | - | Series *B*, composition 2 |
| Fe + 20% Mn₂N | −0.05 | 0.06 | 0.03 | 0.20 | 0.16 | 0.37 | 0.00 | 0.00 | 22 | Series *B*, composition 1 |
| Fe-4.7Cr + 10% Mn₂N | −0.11 | 0.00 | 0.03 | 0.20 | - | - | −0.01 | −0.04 | 24 | Series *B*, composition 4 annealing |
| Fe-9.0N | −0.01 | 0.00 | 0.08 | 0.20 | 0.20 | 0.36 | - | - | - | [30] |
| Fe-8.5N | −0.02 | 0.00 | 0.06 | 0.20 | 0.31 | 0.33 | - | - | - | [31] |

Spectra of/from powders of the initial ferromagnetic alloys Fe–XMn and Fe–YCr, as well as of the α phase in the body of MA samples, present by themselves the sextets with broadened lines which are a superposition of the subspectra *S*(*n*1, *n*2) correspondent to the non-equivalent surroundings of atoms of resonance-exhibiting iron by the impurities of chromium and manganese in the positions (lattice sites) of substitution in the solid solutions with BCC crystal lattice [26–28]. In result of the mechanical alloying with the formation of solid solutions, along with the sextet from the α phase, in the center of the spectra

there appears a broadened asymmetrical singlet from austenite. In a first approximation (without taking into account the difference in the values of Debye–Waller factor of phase components), the numerical ratio of the areas under the integral sextet and central singlet corresponds—and is equal in volume percentages—to the quantitative ratio of the volumes of the $\alpha$ and $\gamma$ phases in the MA alloy.

Quantitative assessment of the nitrogen content $C_N$ (in at.%) in austenite was carried out, similarly to that performed in [15,16,30] under the assumption of repulsive distribution (i.e., mutual repulsion) of nitrogen atoms in a solid solution, by the contribution of the atomic configuration represented by the $D(1)$ doublet (Figure 1) from iron atoms with one nitrogen atom in the nearest octahedral interstitial positions. The value of the integral intensity from the doublet $D(1)$ is related to $C_N$ by the relationship [30]:

$$S_{D(1)} = 6p(1 - p),$$

where $p = C_N(1 - C_N)$ is the fraction of the octahedral interstitial sites in austenite occupied by nitrogen atoms. In order to account for the self-absorption effect, the intensity of $S_{D(1)}$ was extrapolated to the zero thickness of the absorber. The change in the integral intensity of the doublet $D(1)$ can be explained by the self-transfer of nitrogen from nitrides to the solid solution of $\gamma$ phase in the course of deformation-induced dissolution and vice versa during the decomposition of solid solution under conditions of nitrogen binding with chromium or manganese into next-emerging secondary nitrides [15,16].

Calculations of the spectra of/from the ferrite $\alpha$ matrix in the MA alloys and, in particular, of the quantity of impurity of substitution in this matrix were performed in the assumptions that (i) the contributions of chromium and manganese to the change in the isomer shift $I_s$ and effective field $H$ at Fe atoms was additive in nature and (ii) the (binomial) distribution of impurities (namely, Cr and Mn) in the solid solution was of chaotic character [26–28,36]. Estimation of the total content of the chromium and manganese in the solid solution of ferrite matrix of the MA alloy, $\alpha C_{Cr + Mn}$, was performed via establishment of a partial contribution to the spectrum and distribution $p(H)$ from the impurity-free atomic surroundings of resonance-exhibiting iron—i.e., from the sextet $S(n1, n2)$ with $n1 = n2 = 0$, where $n1$ and $n2$ are the quantities (number) of atoms of chromium and manganese in the first and second CS, respectively. The sextet $S(0,0)$ in the diluted Fe–Mn and Fe–Cr solid solutions is easily separated in the form of the peak in the distribution $p(H)$ with magnitudes of fields in the vicinity of 330 kOe, see Figure 2a. Nitrogen in the ferromagnetic component of the MA alloy weakly affects (i) the hyper-fine parameters of the spectrum of $\alpha$ solid solution, as well as (ii) the value of the parameter of the mean effective field $<H>$ [17]. Therefore, we present no quantitative estimations of the nitrogen content in the $\alpha$ solid solution of the MA alloy.

## 3. Mechanical Alloying in the Course of the Ball Milling and Thermal Annealing

### 3.1. Mössbauer Analysis of Mechanical Alloying in the Cases of A and B Series

In the case of samples of the series A: Fe–XMn + 20CrN ($Cr_2N$), the results for MA mixtures in the compositions with X = 0, 6.7, 8.9 wt% Mn are presented in Figure 2 and in Table 1. It is seen that an increase of the concentration of manganese in the initial matrix from 0 to 8.9 wt% leads to the growth of intensity of the central singlet of paramagnetic austenite from 6 to 36 vol.%. In its turn, in the singlet of/from the austenite, the doublets $D(1)$ and $D(2)$ are well pronounced, which testifies that (i) nitrogen «hits» into the octahedral interstitial positions in the crystal lattice of FCC phase and (ii) nitrogen-containing austenite is formed. In the spectrum for the $\alpha$ phase—being a ferrite component of the MA alloy, changes also are observed, namely, there takes place a pronounced lowering in the magnitude of the mean effective magnetic field $<H>$ of the integral sextet at the expense of lowering of a partial contribution to the spectrum and $p(H)$ of the superpositional sextet $S$ ($n1,n2$) with $n1 = n2 = 0$, correspondent to the impurity-free surroundings of iron by the atoms of chromium and manganese [17,18,26,36]. The estimation of the content of impurity of substitution in the MA $\alpha$ solid solution testifies to the increase in the concentration of the

chemical elements of substitution in the matrix of $\alpha$ phase, see the results of BM in Table 1. Thus, based on the spectra results of the change—in result of ball milling (BM)—in the ferrite and austenite components in the samples of series *A*, the chemical elements from the initial nitrides CrN partially pass to the body of the $\alpha$ and $\gamma$ solid solutions.

In the case of samples of the series *B*: Fe–*Y*Cr + 10Mn$_2$N, the results for the MA mixtures are presented in Figure 3 and in Table 2. The spectra for the MA results have qualitatively the same "appearance" as in the case of series *A*. In the sextet that describes a ferromagnetic $\alpha$ solid solution, one can observe the decrease in the magnitude of <*H*> as a consequence of the enrichment of metallic matrix by manganese, i.e., the dissolution of nitrides has taken place and passing of manganese atoms from the nitrides to the $\alpha$ matrix of the MA alloy has occurred in parallel. However, a noticeable quantitative distinction of the experimental results exists relative to those for the series *A*.

This, first of all, concerns the increase (i) in the volume of austenite, as well as (ii) of the degree of alloying of the $\alpha$ phase by manganese. In the case of the samples of the series *B* with the composition formulas 3 and 4 (see Table 2), the ball milling preceding sample compaction brings about the quantity of austenite from 10 to 24 vol.%, whereas in the case of the samples of the series *A* with the composition formulas from 1 to 3 (see Table 1) the processing mentioned entails the increase in the quantity of austenite component from 4 to 7 vol.%. At the same time, the content of alloying elements of substitution in the ferrite matrix, $\alpha C_{Mn+Cr}$, increases in a more pronounced extent in comparison with that one which is characteristic of the case of series *A*. Besides, in the central singlet of/from austenite the component related with the AFM ordering is distinguished to stand obviously apart, see Figure 3a,c. This testifies to the formation in the $\gamma$-phase of the regions enriched in manganese, i.e., having the manganese content greater than that is characteristic of the average for the composition. The values of the parameters for the sextets "typical" of the AFM ordering are presented in Table 3. The content of nitrogen in austenite for the series *A* and *B* amounts from 0.7 wt (2.6 at.) to 3.7 wt (13.1 at.) %.

### 3.2. Mössbauer Analysis of MA Samples after Their Annealing

In the case of samples of the series *A*, the result of an annealing of the MA mixtures at 700 °C consists in the reverse evolution of the spectra with considerable lowering of the intensity of the central singlet of/from austenite, Figure 4a,c. The content of substitution impurity in the $\alpha$ solid solution approaches the initial—or a lower one than that is characteristic of the initial binary Fe–*X*Mn alloy, Table 2 *a*. In particular, in the composition Fe–8.9Mn + 20CrN, after the MA and annealing at 700 °C, for 1 h, the total content of the manganese and chromium in the $\alpha$ phase does not exceed (i) 1.7 wt% at 8.9 wt% Mn in the initial binary alloy and (ii) 10.0 wt% in the MA alloy after BM. The quantity 36 vol.% of austenite remains virtually the same (i.e., unaltered), but the content of nitrogen exhibits a lowering from 1.1 wt (4.2 at.) % to 0.5 wt (1.8 at.) %. An annealing at a higher temperature, at 800 °C, for 1 h, facilitates the occurrence of the $\alpha \rightarrow \gamma$ transition in the samples of compositions 3 and 4 with *X* = 6.7 and 8.9 and makes lower the degree of decomposition of the solid solutions formed in the course of MA. In samples of all the compositions of series *A*, an annealing at a higher temperature preserves the values of <*H*> and of the intensity of *S*(0,0) component of the sextet of/from the $\alpha$ phase from alteration, i.e., the annealing stabilizes the chemical elements of the substitution in the $\alpha$ matrix. The intensity of *D*(1) decreases in a lesser extent, i.e., nitrogen is partly preserved in the solid solution of paramagnetic austenite, see Figure 4b,d and Table 1 *b*.

In the case of series *B*, the annealing of the MA samples differs, first of all, by increasing the quantity of austenite in the cases of all compositions, see Figure 5 and Table 2. After annealing at 800 °C of the samples of composition 5 with *Y* = 8.6 wt%, up to 100% of austenite is formed. In the austenite(-stipulated) singlet of/from the MA alloy one can observe the preservation of the component related to the AFM ordering, which is described by the sextet with the combined magnetic and quadrupole splitting, see Table 3. The same but in a better way as it takes place in the series *A*, annealing at 800 °C stabilizes

the elements of mechanical alloying in the ferrite matrix; the comparison of the results of annealing of samples of the compositions in the series *A* and *B* is shown in Table 2.

### 3.3. TEM Data on the Results of the MA and Subsequent Annealing

According to the TEM data on the MA alloyed powder mixtures *A* and *B*, the same as in the case of mixtures based on the Fe–Ni matrix [18], one observes there the formation of a submicrocrystalline structure (SMC). In Figure 6 the dark-field image and SAED pattern of the powder mixture Fe–8.6Cr + 15Mn$_2$N are presented. The SAED pattern (Figure 6b) contains ring (circular-wise disposed) reflections from greatly misoriented grains of austenite. According to [37], reflections from the nitrides CrN and Cr$_2$N are located near the reflections from austenite. In the dark-field image taken in the reflection «austenite + nitrides» one can observe the images of the nitrides and fragments of austenite. The size of the fragments of matrix (grains and subgrains) amounts to 50–80 nm. Also visible are more disperse (finer) precipitates of size about 2 nm, which correspond to the secondary nitrides of the types CrN and Cr$_2$N.

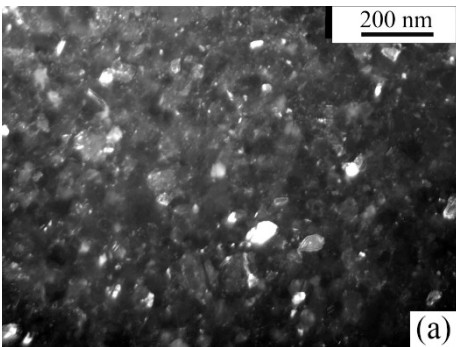
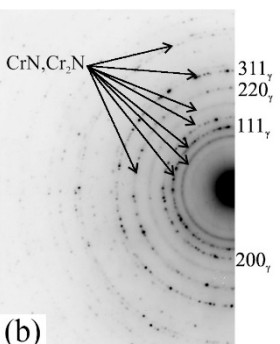

**Figure 6.** TEM structure of the MA powder mixture Fe–8.6Cr + 15Mn2N: (**a**) dark-field image taken in the combined reflection ((111)$_\gamma$ + CrN, Cr$_2$N); (**b**) SAED pattern.

After termination of the annealing at 800 °C for 1 h, an SMC structure of the powder mixture remained unaltered, see Figure 7. The phase composition of the powder mixture Fe–8.6Cr + 15Mn$_2$N presents by itself—after the milling and annealing—the austenite and chromium nitrides (CrN and Cr$_2$N). Annealing of the MA samples leads both to an increase in the size of chromium nitrides, which have been formed in the course of mechanical alloying, and to their precipitation at the dissolution of supersaturated $\gamma$ solid solution. In the SAED pattern shown in Figure 7a, one can see two first reflection rings that correspond to nitrides of chromium (CrN and Cr$_2$N). In the dark-field image taken in the reflection «CrN + Cr$_2$N», (Figure 7b), it is these nitrides that shine.

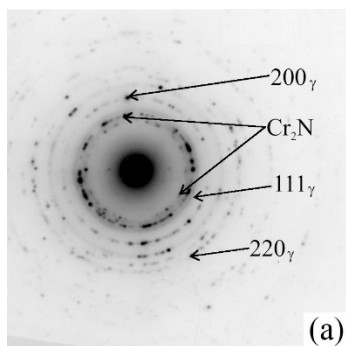
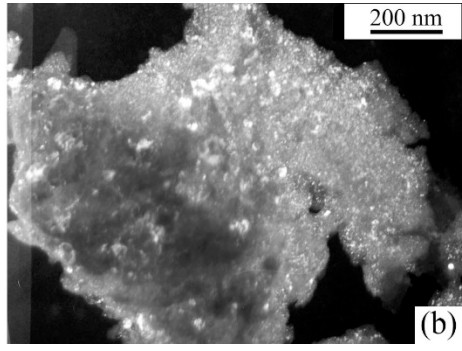

**Figure 7.** TEM structure of the MA powder mixture Fe–8.6Cr + 15Mn$_2$N after annealing at 800 °C for 1 h: (**a**) SAED pattern; (**b**) dark-field image taken in the combined reflection (CrN, Cr$_2$N).

## 4. Discussion

Using as metallic matrix precisely the Fe–Mn and Fe–Cr alloys instead of the pure metals Fe, Mn, and Cr with admixture of nitrides, has its own peculiar features that affect the (i) mechanism of structure-phase transitions, (ii) kinetics, and (iii) properties of the products of MA. This influence is stipulated by the physicochemical properties of the components under alloying and, in particular, by the phase diagram of the alloys and nitrides, as well as (by) the difference in the chemical activity and diffusion mobility of the elements of alloying (Cr, Mn, and N), in the course of the MA and subsequent annealing.

As it follows from the results of Mössbauer investigations, in the samples of series $A$ and $B$ one can observe—under BM—the smelting of the initial binary Fe–Mn and Fe–Cr alloys, on the one hand, and the nitrides CrN and $Mn_2N$, on the other. Experiments on the mechanical alloying in ball mills—used in the capacity of metallic matrix the alloys, —are of common regularities, with SPD-induced (by severe plastic deformation) structure-phase transformations upon the friction- and HPT-providing treatments executed on the nitrided surface of the iron alloys and stainless steels [15–17]. In the course of severe deformation, in MA samples we encounter the formation of solid solutions with the element and phase composition similar to that typical of Fe–Cr–Mn–N alloys produced in a traditional metallurgical way, see data of Tables 1 and 2. The ferromagnetic component having BCC crystal lattice also exhibits a change in its composition by revealing the altering of the quantity of the manganese and chromium in the $\alpha$ solution. In result of an annealing at 700 and 800 °C of the MA samples, both in the series $A$ and in the series $B$, there occurs the $\alpha \to \gamma$ phase transition accompanied by (i) the decomposition of the formed supersaturated solid solutions and (ii) by the exit of nitrogen and of the alloying elements manganese and chromium from the metallic matrices into secondary nitrides, see Tables 1 and 2 $a$, $b$. Moreover, at BM and annealing, in the processes of decomposition of MA solid solutions there participate the chemical elements of an alloying (by chromium and manganese) of the initial binary alloys. After annealing, the content of the chromium and manganese in the $\alpha$ matrix of MA alloy is noticeably decreased in comparison with the content of these elements in the alloys of initial composition. For instance, in the case of samples of composition 4 (Fe–8.9Mn + 20CrN) of $A$ series, the total content of substitution elements in the $\alpha$ phase, $\alpha C_{Mn+Cr}$, decreases to 1.7 wt%, whereas the concentration of nitrogen in austenite does not exceed 0.5 wt (2 at.)%. In the case of series $B$, at 700 °C a qualitatively similar pattern (picture) is observed, see data for the samples of compositions 4 and 5 in Table 2 and in Figure 5a,c. We can assume that active exit of substitution elements from the $\alpha$ matrix is connected not only with the decomposition of supersaturated solid solutions and the formation of nitrides, but also with the diffusion exchange of the chromium and manganese atoms between the $\alpha$ and $\gamma$ phases in accordance with the phase diagrams of these alloys. Moreover, the nanostructurization of metallic matrix in the course of BM contributes to the formation of segregations of the chromium, manganese, and nitrides on along the developed grain boundaries [38,39]. The structure of the nitrogen-containing MA austenite after the thermal annealing and cooling to room temperature experiences a better stabilization precisely after annealing at a higher temperature, namely, of 800 °C, see Tables 1 and 2 $a$, $b$. It can be seen from the Tables that the stabilization of nitrogen-containing austenite at the annealing of higher temperature is explained by the presence of chemical elements of mechanical alloying that have finally remained in the body of a volume of the solid solution.

The data of TEM have confirmed the validity of results of Mössbauer measurements concerning the formation of nitrogen-containing austenite together with the secondary fine nitrides (CrN and $Cr_2N$) in the course of MA of the powder mixtures Fe–Mn–$X$CrN and Fe–Cr–$Y$Mn$_2$N. At high-temperature anneals there occurs both the increase in the size of chromium nitrides that have been formed at mechanical alloying of powder mixtures and the precipitation of new fine ones during the decomposition of supersaturated $\gamma$ solid solution. The preservation at high-temperature annealing of the submicro- and nano-sized austenite grains in MA alloys can be in part explained by the formation of the secondary

nitrides CrN and $Cr_2N$ along the developed grain boundaries, which evolve in parallel of the $\alpha \rightarrow \gamma$ phase transition, slows down the process of recrystallization-controlled growth of grains, see Figure 7a,b.

The alloys obtained by the MA and subsequent annealing at 700 and 800 °C, are metastable and at cooling to room temperature they can pass by the mechanism of shear transformation just to the $\alpha$ phase. The small size of austenite grains [40], short-range-order atomic concentrational separation [41], and precipitation of disperse nitrides coherently matched with the matrix [42], all this play, in the case under consideration, a stabilizing role in relation to the occurrence of the $\gamma \rightarrow \alpha$ transition. An additional stabilization of austenite can be connected with the presence of the concentrational inhomogeneity over a material. The stabilizing role of concentrational inhomogeneities was exemplified on the austenitic Fe–Ni alloys [43]. The formation of the concentrational inhomogeneities over manganese in the austenitic MA samples of series *B*, and more specifically, of the regions of the $\gamma$ phase with an increased content of magnesium, appears both after ball milling and after annealing right in the form of the component related with the AFM ordering, see Figures 1c and 5b. In the spectrum for the alloy Fe–4.7Cr + 10Mn$_2$N annealed at 800 °C, for 1 h, these regions of a structure are described by the sextet with a combined magnetic and quadrupole splitting, see Figure 5b,d and Table 1.

From the results of the BM and anneals of samples from the series *A* and *B*, it follows that the series *B* turns out to be more efficient from the point of view of the volume of a stable austenite being under formation at the lesser (i) quantity *of* alloying elements (*of* chromium, manganese, and nitrogen) and (ii) amount of the primary nitrides in the samples of these compositions, see Table 2. A more efficient formation of the nitrogen-containing austenite and its stabilization in the samples of series *B* can be explained by several reasons.

First, (it can be explained) (i) by a more efficient dissolution of Mn$_2$N nitrides in a metallic matrix and (ii) by the formation of the Mn-rich austenite-forming regions in a metallic matrix of MA alloy with the concentration of manganese ~20 at. %. Manganese-rich regions of austenite can be formed by the actual smelting of nitrides both with austenite and with a paternal $\alpha$ phase. A more active alloying of the initial matrix in the samples of series *B* is well illustrated by data from Tables 1 and 2. The manganese-rich regions of the $\alpha$ phase in this very case possess an enhanced stimulus to/for the $\alpha \rightarrow \gamma$ transition in accordance with the phase diagram (in) [21,22]. Accommodation stresses between (i) the $\gamma$ phase enriched in manganese during its formation in the course of MA treatment and (ii) the rest matrix, stabilize austenite in relation to the $\gamma \rightarrow \alpha$ transition [42,43].

Second, the high efficiency of the formation of austenite in the samples of series *B* can be explained by a low diffusion mobility of the atoms of manganese (having a larger atomic radius in comparison with that of chromium), which (i) makes weaker the process of dynamic aging (at BM) concurrent with the process of dissolution of nitrides and (ii) facilitates the preservation of manganese to remain in the solid solution. The (in-case) presence of the dynamic aging concurrent with the dissolution is obvious from the slight chromium doping of the initial Fe–*X*Mn matrix, see the result of BM in Table 1 that presents data for the case of series *A*. The accelerated concurrent process in the samples of series *A* presumably is explained by the enhanced chemical and diffusion activity of chromium. It is noticeable that it is in the case of series *B* there more actively occur both dissolution of Mn$_2$N nitrides and alloying of the $\alpha$ phase by manganese, see Table 2. The formation of the regions enriched in manganese is most distinctly revealed in conditions of the BM at relatively moderate temperatures. Synergetic nature of deformation-induced processes, which manifests itself in the decomposition of supersaturated solid solutions in the course of concurrent mechanical alloying, was demonstrated in the works on the short-range order investigations into the Fe–Ni and Fe–Cr alloys [44,45]. The influence of the diffusion mobility of the alloying elements of substitution on this process was shown in the works on the dynamic aging in the Fe–Ni–*Me* alloys where *Me* = Ti, Al, Si, Zr [44,46,47].

The third reason for the high efficiency of the formation of austenite in the case of series *B* consists in the lower thermal stability of $Mn_2N$ nitrides in comparison with that of the nitrides CrN ($Cr_2N$) in conditions of the local heating-up of the mixture at the BM and high-temperature anneals. According to the data from [48], the nitride $Mn_2N$ undergoes decomposition at 900 °C, while the nitride CrN decomposes at sufficiently higher temperatures (>1000 °C). The effect of the thermal stability and chemical activity of the constituents of would-be MA products on the kinetics of the dynamic aging and formation of secondary phases was demonstrated in the example of deformation-induced dissolution of nitrides, as well as intermetallics in [44,46,47].

## 5. Conclusions

Using the methods of Mössbauer spectroscopy and electron microscopy, we have investigated the structure-phase transitions in the course of solid-phase mechanical alloying in a ball mill and at subsequent annealing of the mixtures of the Fe–Cr and Fe–Mn alloys, on the one hand, and the nitrides CrN ($Cr_2N$) and $Mn_2N$, on the other. The samples from the series *A*, of compositions Fe–XMn + 20CrN ($Cr_2N$), and the samples from the series *B*, of compositions Fe–YCr + 10, 20$Mn_2N$, —were the objects of investigation. It is shown that the ball milling of the mixtures from the series *A* and *B* leads to the smelting of the components of mixtures and to the formation of the Fe–Cr–Mn–N solid solutions with FCC and BCC crystal lattices. Increasing the concentration of the elements Cr and Mn in the initial BCC alloys, as well as the proportion amount of nitrides in the studied compositions in conditions of the ball milling and subsequent annealing, is accompanied by the increase in the volume of forming austenite with nitrogen concentration of 3.7 wt (13.1 at.)%. A result of the mechanical alloying of samples from the series *B* is the formation of the FCC phase, part of which is of AFM ordering development with the content of manganese equal or greater ($\geq$) than 20 wt%.

The process of the ball milling and subsequent isothermal annealing of the MA mixtures correspondent to the series *A* and *B* is accompanied by the $\alpha \rightarrow \gamma$ phase transitions, formation of nitrogen-supersaturated solid solutions, and their decomposition with the formation of secondary nitrides. Austenite formed in the course of ball milling and subsequent annealing at 700 and 800 °C, for 1, has submicro- and nano-sized grains, which along with the coherent disperse (fine) secondary nitrides and alloying elements are preserved in the matrix, which is a reason for the stabilization of austenite. It has been established experimentally that annealing of MA samples at 700 °C leads to a sharp decrease in the amount of the alloying elements of manganese and chromium in the $\alpha$ phase, which along with the decomposition and formation of secondary nitrides can be a consequence of diffusion of chromium and manganese from the volume of the $\alpha$ into $\gamma$ phase. It is shown that the *B* series—in its samples with the primary nitrides $Mn_2N$—is both at ball milling and at subsequent annealing more efficient from the viewpoint of forming the greater volume of MA austenite and its saturation with nitrogen at a lesser—than is characteristic of the case of the series *A*—content of the alloying chromium and manganese. The higher stability of MA austenite in the case of the series *B* is explained by the (i) formation of concentration-inhomogeneous regions of the matrix enriched in austenite-forming manganese, (ii) thermal stability of $Mn_2N$ nitrides being lower in comparison with that of CrN ($Cr_2N$), and (iii) low diffusion mobility of manganese in the course of the ball milling and subsequent annealing.

**Author Contributions:** Conceptualization, V.S.; methodology, V.S., E.N., K.L., A.Z., N.K. and K.K.; validation, V.S., K.L., N.K. and K.K.; formal Analysis, V.S., A.Z., N.K., K.L. and K.K.; investigation, K.L., K.K. and E.N.; writing—original draft preparation, V.S. and K.L.; writing—review and editing, Y.U., K.L. and K.K.; project administration, V.S.; funding acquisition, K.L. and V.S. All authors have read and agreed to the published version of the manuscript.

**Funding:** The reported study was funded by the Russian Foundation for Basic Research (project no. 19-33-60006). The research was carried out within the state assignment of Ministry of Science and Higher Education of the Russian Federation (theme "structure" no. AAAA-A18-118020190116-6.

**Acknowledgments:** The authors are grateful to Gennady Dorofeev for the providing of nitride powders and their attestation.

**Conflicts of Interest:** The authors declare no conflict of interest.

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
