# Peer review of "Structure-Phase Transformations in the Course of Solid-State Mechanical Alloying of High-Nitrogen Chromium-Manganese Steels"

_metals, doi:10.3390/met11020301_

Round 1
Reviewer 1 Report
The submitted paper deals with elaboration of Fe-Cr-Mn, N-rich austenitic steels. A mechanical alloying route of their preparation is chosen and the obtained powders are analysed mainly by Mössbauer spectroscopy (MS), associated with TEM local analyses. The topic is interesting. Unfortunately, in the present shape, the paper is not acceptable for publication in Metals. Several main reasons of such Reviewer’s opinion can be raised.
- About the scientific content.
The way of conducting the study, the way of the results presentation and their analysis are to be questioned in several points.
- The materials are prepared by mechanical alloying. Contamination (ball material, oxygen, nitrogen …) is a current issue in this kind of process. It is necessary to discuss this topic or at least mention it.
- Also, in the case of ball milling: what is the material of balls used? the expected contamination of powder will depend on it.
- Materials chemical composition is given only as a “nominal” one: results of chemical compositions measurements of obtained materials are necessary to confirm the results and reinforce the authors’ conclusions.
- The analyses of chemical composition of identified phases is a big issue in the presented data. Especially, in 3.2.: which way the evolution of chemical composition was determined and quantified? The results are interesting, but not sufficiently evidenced.
- And also: the electron diffraction patterns (SA, TEM) are not sufficiently analysed. What is the crystallographic structure, and lattice parameters of identified phases, especially: the nitrides ? Why {111} ring of austenite is not seen/shown in fig. 7a ? why ferrite is not observed in any pattern ? how is it possible, by SA method, to distinguish austenite from nitrides in dark field (fig. 7b) ?
- About the paper presentation
- The introduction is too short and doesn’t cover the whole field of study. Thus, it doesn’t provide the lecturer with the “state-of-the-art” knowledge necessary to understand the interest of the paper. Especially, a review of existing data on Mössbauer spectroscopy (MS) use to evaluate the phase transformations, as far as possible: in steels, in the presence of interstitial elements (C, N) would be appreciated.
- Mössbauer spectroscopy is not a really current experimental technique. It gives very rich information about atomic-scale structure of phases. To help the lecturer to appreciate the strong points of MS in the study, a comprehensive review of expected information is necessary. When (and why ?) a magnetic sextet is observed (also : line154, its presence is associated to the austenitic phase : is there a contradiction with a paramagnetic behavior of FCC austenite in steels ?) ? what information can be obtained from the measured isomer shift ? etc. …
- It is surprising to see major part of references coming from very old papers (1970-1990). The ref. [14] (phase diagrams, 1986) should be replaced or at least completed by more recent papers. Almost all recent references are self-citations.
- A presentation of results starting with an overall view of the obtained powders (their X-ray diffraction analyses for phase identification and chemical analyses results for obtained powders), followed by TEM analyses to confirm the identified structures and preparing the MS results presentation that allow the comprehension of phenomena involved during MA, would be easier to follow and thus, more efficient to convince the reader.
- Finally, some conclusions in discussion part seem not be sufficiently supported by the obtained results. For instance : lines 247-255 : the detailed analyses of evolution of chemical compositions of phases upon annealing is not supported by identified analyses ; line 312 ; the kinetics of transformations is not really studied : lines 331-332 : the conclusion is unclear ; line 344-345 : there is no evidence of such segregation in this work ; line 364 : no evidence of SRO phenomena in the studied alloys ; the reference is a self-citation of a work dealing with ordered FeNi systems ; lines 390-391 : no evidence, or reference, for slower Mn diffusion as compared to Cr is given ; lines 434-438 : no experimental evidence of “sharp decrease of alloying elements” (…) has been seen
- About the English level and, generally, the quality of written expression.
- A detail: ‘”mechanical alloying” (instead of “mechanical synthesis”) is a better, currently admitted term to describe the process used in this work.
- The sentences are often much too long (eg.: lines 172 to 176), and often, unclear.
- The English syntax needs to be verified and improved.
- Very often, some words are put into the brackets or parenthesis: it is not convenient. If this part of a sentence is necessary, the brackets are not necessary; otherwise, these “words” are not necessary. Many examples exist: line 30, 82, 100, … 350, 351.
- An in-depth analysis of the whole text is inavoidable. An English-native speaker’s help would be strongly recommended.
Author Response
Dear Sir,
Thank you for your close peer-reviewing. It helped us exclude many mistakes.
The authors have taken into account all remarks and performed corrections, having marked these corrections in the text by a yellow color.
- About the scientific content.
Q:
- The materials are prepared by mechanical alloying. Contamination (ball material, oxygen, nitrogen…) is a current issue in this kind of process. It is necessary to discuss this topic or at least mention it.
- Also, in the case of ball milling: what is the material of balls used? the expected contamination of powder will depend on it.
A:
Performed.
The following Information on the problem of contamination has been added to the text:
“The vessel and balls were made of high-strength ball-bearing steel containing 1.5 wt % Cr and 1.0 wt % C, iron the balance. This steel is the most stable compared to materials such as tungsten carbide and stainless steel.
After termination of ball milling, part of the powders were annealed in vacuum at temperatures of 700 and 800°C for 1 hour, to specify – in a vacuum of 10-5 mm Hg. Possible contamination of the test samples with wear products was controlled by weighing the mass of vessels, balls, and powder before and after mechanical fusion–alloying. The difference in the masses of powders did not exceed 0.3-0.5 wt%. According to the results of chemical analysis, in all the cases of ball milling, the oxygen concentration in the MA samples did not exceed 0.8 wt%.” (lines 86–99 in revised version)
Q:
- Materials chemical composition is given only as a “nominal” one: results of chemical compositions measurements of obtained materials are necessary to confirm the results and reinforce the authors’ conclusions.
- The analyses of chemical composition of identified phases is a big issue in the presented data. Especially, in 3.2.: which way the evolution of chemical composition was determined and quantified? The results are interesting, but not sufficiently evidenced.
A:
The phase and chemical composition were analyzed by the MS method separately for the a and g phases on the basis of the (i) methods described in 2.2, as well as (ii) apriori information on the control of the nitrogen content in austenite and manganese and the chromium content in ferrite of the MA samples obtained. For verification, in addition, model MS experiments on the composition of MA samples were performed. Information on the assessment of nitrogen in austenite by the assessment of the D1 doublet intensity has been added to the text. The results of the evolution of the composition are shown in Tables 1 and 2. The results of the analysis do not contradict the data of the equilibrium and non-equilibrium phase diagrams of Fe-Cr-Mn-N alloys [21,22].
The following Information on the problem of contamination has been added to the text:
“Quantitative assessment of the nitrogen content CN (in at.%) in austenite was carried out – similarly to that performed in [16,30] under the assumption of repulsive distribution (i.e., mutual repulsion) of nitrogen atoms in a solid solution – by the contribution of the atomic configuration represented by the D(1) doublet (Figure 1) from iron atoms with one nitrogen atom in the nearest octahedral interstitial positions. The value of the integral intensity from the doublet D(1) is related to CN by the relationship [21]: D(1) = 6p(1 – p),
where p = CN(1 – CN) is the fraction of the octahedral interstitial sites in austenite occupied by nitrogen atoms. In order to account for the self-absorption effect, the intensity of SD(1) was extrapolated to the zero thickness of the absorber.” (lines 189–197 in revised version)
Q:
And also: the electron diffraction patterns (SA, TEM) are not sufficiently analysed. What is the crystallographic structure, and lattice parameters of identified phases, especially: the nitrides? Why {111} ring of austenite is not seen/shown in fig. 7a? why ferrite is not observed in any pattern? how is it possible, by SA method, to distinguish austenite from nitrides in dark field (fig. 7b)
A:
Performed:
The ring 111g now is shown in Fig.7a.
Electron microscopic studies were performed on the alloy (Fe–9Cr+15%Mn2N) which is a single-phase austenitic alloy. There is no ferrite in this alloy.
The dark-field image (Fig.7b) was taken in the reflections that are correspondent to the phase Cr2N.
In the dark-field image, austenite is different from nitrides of chromium morphologically. If austenite had been in the reflective position, then austenite would have shined (been illuminated) on the micrograph. In the presented image, disperse (fine) particles of chromium nitrides are illuminated against the background of a dark austenitic matrix.
For the identification of the diffraction patterns we made use of the data of Output generated by Single Crystal (TM) for Windows 2.3.2 Copyright © 1994-2015 Crystal Maker Software Ltd http://www.crystalmaker.com.
- About the paper presentation
Q:
The introduction is too short and doesn’t cover the whole field of study. Thus, it doesn’t provide the lecturer with the “state-of-the-art” knowledge necessary to understand the interest of the paper. Especially, a review of existing data on Mössbauer spectroscopy (MS) use to evaluate the phase transformations, as far as possible: in steels, in the presence of interstitial elements (C, N) would be appreciated.
A:
Performed. Added the following:
“The current technology for melting high-nitrogen steels under nitrogen pressure requires expensive equipment and high energy expenditure. In the capacity of an alternative and cheaper method for producing high-nitrogen steels, today powder metallurgy methods using mechanical alloying are currently proposed. In most studies on the MA of high-nitrogen steels, the metal powder was saturated with nitrogen from the gas phase in a flowing atmosphere of nitrogen or ammonia [4–13]. In these works, the authors usually employed vibrationally-assisted ball milling with long-term processing times (100 hours or more). In the works [14–20], for the purpose of nitriding Fe-based steels, the authors were the first to have employed the method of solid-phase MA (i.e., MA of materials in a solid state). This method does not require additional gas equipment to create a flowing nitrogen-containing atmosphere in the mill vessel. In addition, the method does not require such a long time of mechanical processing and allows for one to use planetary-type ball mills”. (lines 33–44 in revised version)
Q:
Mössbauer spectroscopy is not a really current experimental technique. It gives very rich information about atomic-scale structure of phases. To help the lecturer to appreciate the strong points of MS in the study, a comprehensive review of expected information is necessary. When (and why?) a magnetic sextet is observed (also: line154, its presence is associated to the austenitic phase is there a contradiction with a paramagnetic behavior of FCC austenite in steels?) ? what information can be obtained from the measured isomer shift ? etc. …
A:
In this paper, the method of analysis of the chemical and phase composition at MA is the MS method. We took for the investigation, in the capacity of initial materials, the Fe-Mn and Fe-Cr alloys, as well as the nitrides CrN(Cr2N) and Mn2N, which all were certified by the chemical composition; see section 2.1 and Tables 1 and 2.
The interpretation of the results is non-controversial and quite consistent with the Mössbauer data on the phase and concentration composition of Fe-Cr-Mn-N solid solutions.
Data on the formation of a magnetic sextet follow from the magnetic phase diagram of Fe-Mn alloys.
Q:
It is surprising to see major part of references coming from very old papers (1970-1990). The ref. [14] (phase diagrams, 1986) should be replaced or at least completed by more recent papers. Almost all recent references are self-citations.
A:
There is a large number of works on Mössbauer spectroscopy on the study of nitrogenous austenite and ferrite obtained by metallurgical means and under conditions of thermomechanical treatments. The main necessary works related mainly to the 80 and 90 years are cited. In accordance with the reviewer's recommendation, we removed the reference [14] and added to the text of the work several references to the works on mechanical fusion–alloying of high-nitrogen steels, which were performed using vibrationally-assisted ball mills and a gas medium.
Q:
A presentation of results starting with an overall view of the obtained powders (their X-ray diffraction analyses for phase identification and chemical analyses results for obtained powders), followed by TEM analyses to confirm the identified structures and preparing the MS results presentation that allow the comprehension of phenomena involved during MA, would be easier to follow and thus, more efficient to convince the reader.
A:
The interpretation of the experimental results in the section DISCUSSION, in particular, on the temperature stabilization of the structure of MA alloys, is based on the known structural and diffusion data for the alloys under investigation of different phase composition, morphology and obtained under conditions of heavy plastic deformation.
In this paper, the evolution of the chemical composition of phases is analyzed by the method of Mössbauer spectroscopy. Other analysis is problematic.
Finally, some conclusions in discussion part seem not be sufficiently supported by the obtained results. For instance:
Q:
– lines 247-255: the detailed analyses of evolution of chemical compositions of phases upon annealing is not supported by identified analyses;
– line 312: the kinetics of transformations is not really studied:
–lines 331-332: the conclusion is unclear;
A:
The kinetics of transformations follows from the data in Tables 1 and 2. The conclusion is given in detail in the section DISCUSSION (lines 337-400 in revised version).
Q:
line 344-345: there is no evidence of such segregation in this work;
A:
The aim of the work is not to detect segregations, but the possibility of their formation is justified in the literature, and therefore, along with the reference [38], an additional reference [39] is added to the list of cited literature.
Q:
line 364: no evidence of SRO phenomena in the studied alloys; the reference is a self-citation of a work dealing with ordered FeNi systems;
A:
It's not really obvious, but it's possible. This conclusion is made in accordance with the data of works on SRO phenomena in austenitic Fe-Ni alloys.
Q:
lines 390-391: no evidence, or reference, for slower Mn diffusion as compared to Cr is given;
A:
The assumption of a slower diffusion of Mn involved in the decomposition of solid solutions follows from the data in Tables 1 and 2, as well as the known relationship of diffusion mobility with the size of atoms. In this case, the talk is about the atoms of Mn, which have an increased size of the atomic radius.
Q:
lines 434-438: no experimental evidence of “sharp decrease of alloying elements” (…) has been seen.
A:
Data on " sharp decrease of alloying elements” are shown in Tables 1 and 2 (see for example lines 3 and 4 in Table 1 and lines 4 and 5 in Table 2).
- About the English level and, generally, the quality of written expression.
Q:
A detail: ‘”mechanical alloying” (instead of “mechanical synthesis”) is a better, currently admitted term to describe the process used in this work.
A:
Performed
Q:
The sentences are often much too long (eg.: lines 172 to 176), and often, unclear.
A:
Performed
Q:
The English syntax needs to be verified and improved.
A:
Performed
Q:
Very often, some words are put into the brackets or parenthesis: it is not convenient. If this part of a sentence is necessary, the brackets are not necessary; otherwise, these “words” are not necessary. Many examples exist: line 30, 82, 100, … 350, 351.
A:
Performed
Q:
An in-depth analysis of the whole text is inavoidable. An English-native speaker’s help would be strongly recommended.
A:
Performed
With best regards,
the authors

Reviewer 2 Report
The investigation on the new systhesis technology of Fe-Mn-Cr-N grades are innovative and good reference for researchers. The methods and results are clearly demonstrated. It is suggested the authors re-formulate the long sentences for easy understanding of the audience:
- remove most of the content in (), and embedded them in short sentences.
- improve language, grammar mistakes, for example "5. conclusion" should be "5. conclusions"
- It would be helpful to provide the size distribution of the precipitates in the TEM figures in Figure7. It seems more fine precipitates are exisiting after heat treatment. So the audience can learn more about the heat treatment on the precipitation development.
Author Response
Dear Sir,
We thank you for your remarks.
The authors have taken into account your remarks and performed corrections, having marked these corrections in the text by a yellow color.
Comments and Suggestions for Authors.
Q:
remove most of the content in (), and embedded them in short sentences.
A:
If possible, we have completed.
Q:
improve language, grammar mistakes, for example "5. conclusion" should be "5. conclusions".
A:
Performed:
Q:
It would be helpful to provide the size distribution of the precipitates in the TEM figures in Figure7. It seems more fine precipitates are exisiting after heat treatment. So the audience can learn more about the heat treatment on the precipitation development.
A:
The proposal is valid, but the separation of primary and secondary particles is a complex task and requires the involvement of additional methods of analysis by size, morphology, etc. Such work requires special setting of a problem.
The aim of this work was to establish the kinetic characteristics induced by the deformation of structural-phase transformations of mechanical doping and stabilization of the formed MA structure.
With best regards,
the authors

Reviewer 3 Report
Mechanical synthesis (grinding in a mill) became very interesting, moreover for large industry production of materials. Also Mössbauer spectroscopy has a big potential in the industry applications, even is not still fully used in common industrial use.
In my opinion the presented manuscript is important in such field of research of iron alloys. Mechanical synthesis and prepared samples composition are well described in detail. Mössbauer analysis and spectra interpretation are deeply discussed.
Real spectra analysis and their comparison with simulated spectra gives the credibility of the methods selected.
But, what is the quality of the Mössbauer data?
Comment 1: Are the presented spectra (experimental points) the exact raw data, or processed by some “filtration” since authors discussed the SW for poor spectra resolution in line 102? Please mention in the paper this point if it is true – preprocessing was applied or not. If yes, please present some “raw data” spectrum.
Nevertheless, in my opinion the Mössbauer spectra in presented form are as pictures, not spectra or graphs, since Y-scale is missing (real and/or filtered/simulated?). In my opinion y-scale should be in “counts” units, accompanied, if preferred, by resonance effect (%) unit.
Comment 2: Please include the y-scale to all Mössbauer spectra.
The same should be processed in the case of p(V) and p(H) distributions, where more differences in the probability will arise.
Comment 3: Please include the y-scale to p(V) and p(H) graphs.
Comment 4: Tables 1 and 2 present results of XRD measurements by TEM? Please specify in captions, if yes. Also, what is the XRD uncertainty of the determination of % for presented quantities?
Comment 5: Can authors present typical XRD spectra for selected samples?
Comment 6: Results of both XRD and Mössbauer analyses should be compared even from the point of measurement precision, i.e. amount of austenite content.
Comment 7: In figure 3 p(V) instead of p(H) on y-scale labels should be in simulated distributions.
Comment 8: In table 3 uncertainties for all parameters are missing. Please mention it. Also can you declare why IS and QS are calculated with different precisions? I.e. for reviewer only, if preferred.
According to the experimental data (points) readers should to evaluate the fit (analysis) precision for all the Mössbauer spectra. Then number of spectral components can be discussed, by reader.
In my opinion, while the manuscripts deals mainly with Mössbauer analysis, more detailed information should be mentioned.
Comment 10: Following the Comment 6, what is the uncertainty of determination spectral components in Mössbauer spectra?
Comment 11: What is the number of velocity channels of Mössbauer spectra (512, 1024...)? For intervals approx. +/- 10 mm/s – this will affect the precision of hyperfine parameters determination, in combination of spectra processing by fitting SW. Please specify.
Comment 12: Spectra with small velocity intervals +/- 2 mm/s were measured also with the same number of channels (512, 1024...), or not? Precision should be much better.
Comment 13: In figures 2 and 3 the details of spectra central parts are re-measured at low velocity intervals? It looks like it. Or it is zoomed. Please specify.
Comment 14: In line 136 is “... + D(1) + D(2) + SAFM with the parameters given in Table 1. The selection of the Gaussian forms for the ...” in Table 3 should be.
Comment 15: In table 3 hyperfine parameters of only simulated spectra are presented? Please specify in caption if yes. If yes, the presence of small sextets (22-24 kOe) in real spectra is not evident. Agree? Then comparison of simulated and real spectra hyperfine parameters (for each component) should be presented – readers can evaluate the model. In my opinion after reading, it is not so clear if simulated or real spectra are analyzed/discussed.
Comment 16: In lines 163-165 is discussed Fe substitution by Cr and Mn in the BCC lattice. Later main last peak in p(H) distributions is assigned to S(0,0) sextet, alpha iron. Can authors assign the other peaks of p(H) to relevant Cr/Mn substitutions of Fe. On this data, when y-axis scale will be presented, one can discuss the amount of Cr or Mn occurrence in the BCC lattice. In my opinion, this analysis will be helpful for readers, to demonstrate the power of the Mössbauer spectroscopy, also. It will be beneficial to include it in this study, where samples were prepared by the well-known process. It can be discussed with the author statements as “...low diffusion mobility of the atoms of manganese (in comparison with that of chromium), ... enhanced chemical and diffusion activity of chromium...” on lines 391 and 397.
Conclusion:
Again mentioned, I think, the paper can by very valuable even for Mössbauer community, but more detailed technical and experimental parameters should be presented, since all the manuscript results are covered by Mössbauer spectroscopy mainly, with some method-limits.
Author Response
Dear Sir,
We thank you for your remarks.
The authors have taken into account all remarks and performed corrections, having marked these corrections in the text by a yellow color.
Comments and Suggestions for Authors.
Q:
– Comment 1: Are the presented spectra (experimental points) the exact raw data, or processed by some “filtration” since authors discussed the SW for poor spectra resolution in line 102? Please mention in the paper this point if it is true – preprocessing was applied or not. If yes, please present some “raw data” spectrum.
Nevertheless, in my opinion the Mössbauer spectra in presented form are as pictures, not spectra or graphs, since Y-scale is missing (real and/or filtered/simulated?). In my opinion y-scale should be in “counts” units, accompanied, if preferred, by resonance effect (%) unit.
– Comment 2: Please include the y-scale to all Mössbauer spectra.
The same should be processed in the case of p(V) and p(H) distributions, where more differences in the probability will arise.
– Comment 3: Please include the y-scale to p(V) and p(H) graphs.
A:
In section 2.2 we added information about source of the Mössbauer spectra and the results of their calculation:
Also the following Information has been added to the text:
“The Mössbauer spectra (transmission of gamma-quanta A, % as a function of Doppler velocity V, mm/s) of MA samples have revealed in themselves a multicomponent structure typical of the solid solutions in both α-ferrite and γ-austenite phase states” (lines 112–114 in revised version)
and “Figures 1–5 show the initial experimental Mössbauer spectra and the results of their calculation” (lines 116–117)
The reviewer's recommendation concerning “Figures with the addition of transmission of gamma-quanta scales A, %, and distributions p(H) and p(V), r.u.” was fulfilled.
Q:
Comment 4: Tables 1 and 2 present results of XRD measurements by TEM? Please specify in captions, if yes. Also, what is the XRD uncertainty of the determination of % for presented quantities?
Comment 5: Can authors present typical XRD spectra for selected samples?
Comment 6: Results of both XRD and Mössbauer analyses should be compared even from the point of measurement precision, i.e. amount of austenite content".
A:
The Tables show the Mössbauer data on changes in the N content in austenite and Mn + Cr content in ferrite.
Section 2.2 adds information in the form of a formula for determining the nitrogen content in austenite from the partial contribution of D1 to the austenite singlet:
Q:
Comment 7: In figure 3 p(V) instead of p(H) on y-scale labels should be in simulated distributions.
A:
Performed.
Q:
Comment 8: In table 3 uncertainties for all parameters are missing. Please mention it. Also can you declare why IS and QS are calculated with different precisions? I.e. for reviewer only, if preferred.
A:
Performed.
The reviewer's remark is true. The error is indicated by the results of a computer calculation (see Table 3).
Q:
Comment 9: According to the experimental data (points) readers should to evaluate the fit (analysis) precision for all the Mössbauer spectra. Then number of spectral components can be discussed, by reader.
In my opinion, while the manuscripts deals mainly with Mössbauer analysis, more detailed information should be mentioned.
Comment 10: Following the Comment 6, what is the uncertainty of determination spectral components in Mössbauer spectra?
A:
The phase and chemical composition were analyzed by the MS method separately for the a and g phases on the basis of the (i) methods described in 2.2, as well as (ii) apriori information on the control of the nitrogen content in austenite and manganese and the chromium content in ferrite of the MA samples obtained. For verification, in addition, model MS experiments on the composition of MA samples were performed. Information on the assessment of nitrogen in austenite by the assessment of the D1 doublet intensity has been added to the text. The results of the evolution of the composition are shown in Tables 1 and 2. The results of the analysis do not contradict the data of the equilibrium and non-equilibrium phase diagrams of Fe-Cr-Mn-N alloys [21,22].
Quantitative assessment of the nitrogen content CN (in at.%) in austenite was carried out – similarly to that performed in [16,30] under the assumption of repulsive distribution (i.e., mutual repulsion) of nitrogen atoms in a solid solution.
Q:
Comment 11: What is the number of velocity channels of Mössbauer spectra (512, 1024...)? For intervals approx. +/- 10 mm/s – this will affect the precision of hyperfine parameters determination, in combination of spectra processing by fitting SW. Please specify
Comment 12: Spectra with small velocity intervals +/- 2 mm/s were measured also with the same number of channels (512, 1024...), or not? Precision should be much better.
Comment 13: In figures 2 and 3 the details of spectra central parts are re-measured at low velocity intervals? It looks like it. Or it is zoomed. Please specify.
A:
Measurements in the range of a - Fe were carried out on 512 channels.
Measurements in the range of sodium nitroprusside were carried out on 256 channels.
Q:
Comment 14: In line 136 is “... + D(1) + D(2) + SAFM with the parameters given in Table 1. The selection of the Gaussian forms for the ...” in Table 3 should be.
A:
The values of hyperfine parameters from the calculation of p(V) obtained by approximation with Gaussian forms were the initial data for the subsequent iteration in the model of the superposition of Lorentz forms with a small interval of changes.
The direct solution problem is unstable. In this regard, the authors do not consider information about the results of approximation by Gaussian forms mandatory for presentation.
Q:
Comment 15: In table 3 hyperfine parameters of only simulated spectra are presented? Please specify in caption if yes. If yes, the presence of small sextets (22-24 kOe) in real spectra is not evident. Agree? Then comparison of simulated and real spectra hyperfine parameters (for each component) should be presented – readers can evaluate the model. In my opinion after reading, it is not so clear if simulated or real spectra are analyzed/discussed.
A:
We agree.
In the caption to Table 3, information is added that the parameters of simulated spectra are given.
Q:
Comment 16: In lines 163–165 is discussed Fe substitution by Cr and Mn in the BCC lattice. Later main last peak in p(H) distributions is assigned to S(0,0) sextet, alpha iron. Can authors assign the other peaks of p(H) to relevant Cr/Mn substitutions of Fe. On this data, when y-axis scale will be presented, one can discuss the amount of Cr or Mn occurrence in the BCC lattice. In my opinion, this analysis will be helpful for readers, to demonstrate the power of the Mössbauer spectroscopy, also. It will be beneficial to include it in this study, where samples were prepared by the well-known process. It can be discussed with the author statements as “...low diffusion mobility of the atoms of manganese (in comparison with that of chromium), ... enhanced chemical and diffusion activity of chromium...” on lines 391 and 397.
A:
The reviewer suggests an interesting problem, but with a given doping by Mn and Cr, it may be difficult to implement due to the proximity of the values of DH1 and DH2 of the doping elements.
With best regards,
the authors

Round 2
Reviewer 3 Report
I would like to thank to authors for additional work on manuscript, which is in publishable form now, in my opinion.
Just to check, if in the line 152 Table 3 should be referred instead of Table 1.
Author Response
Dear Sir,
We are very grateful to You for Your carefully reading and peer-reviewing of our work. It helped us exclude many mistakes. We have performed correction concerning your remark.
Yours truly,
the authors